# Fabrication of Polymer Microfluidics: An Overview

**DOI:** 10.3390/polym14102028

**Published:** 2022-05-16

**Authors:** Yi-Je Juang, Yu-Jui Chiu

**Affiliations:** 1Department of Chemical Engineering, National Cheng Kung University, No. 1 University Road, Tainan 70101, Taiwan; n36101284@gmail.com; 2Core Facility Center, National Cheng Kung University, No. 1 University Road, Tainan 70101, Taiwan; 3Research Center for Energy Technology and Strategy, National Cheng Kung University, No.1 University Road, Tainan 70101, Taiwan

**Keywords:** polymer microfluidics, micro-embossing, micro-injection molding, PDMS casting, CNC micromachining, laser micromachining, 3D printing

## Abstract

Microfluidic platform technology has presented a new strategy to detect and analyze analytes and biological entities thanks to its reduced dimensions, which results in lower reagent consumption, fast reaction, multiplex, simplified procedure, and high portability. In addition, various forces, such as hydrodynamic force, electrokinetic force, and acoustic force, become available to manipulate particles to be focused and aligned, sorted, trapped, patterned, etc. To fabricate microfluidic chips, silicon was the first to be used as a substrate material because its processing is highly correlated to semiconductor fabrication techniques. Nevertheless, other materials, such as glass, polymers, ceramics, and metals, were also adopted during the emergence of microfluidics. Among numerous applications of microfluidics, where repeated short-time monitoring and one-time usage at an affordable price is required, polymer microfluidics has stood out to fulfill demand by making good use of its variety in material properties and processing techniques. In this paper, the primary fabrication techniques for polymer microfluidics were reviewed and classified into two categories, e.g., mold-based and non-mold-based approaches. For the mold-based approaches, micro-embossing, micro-injection molding, and casting were discussed. As for the non-mold-based approaches, CNC micromachining, laser micromachining, and 3D printing were discussed. This review provides researchers and the general audience with an overview of the fabrication techniques of polymer microfluidic devices, which could serve as a reference when one embarks on studies in this field and deals with polymer microfluidics.

## 1. Introduction

Microfluidics is the science and technology of systems that process or manipulate small amounts of fluids and particles in the channels with dimensions of tens to hundreds of micrometers [1]. With the ongoing extensive research and development of microfluidic platform technology, its potential applications have been constantly explored and demonstrated, ranging from chemical and biological detection and analysis [2,3], the synthesis and characterization of catalyst particles [4], point-of-care diagnoses [5], drug discovery and delivery systems, food safety inspection [6], environmental monitoring [7,8], life sciences [9], and energy generation [10,11,12,13]. Due to its small scale, microfluidics presents competitive advantages over conventional approaches, such as lower reagent consumption and waste, enhanced reaction efficiency, reduced analysis time, simplified procedures, and high portability [14]. In addition, there are various forces, such as hydrodynamic force, electrokinetic force, acoustic force, magnetic force, centrifugal force, and capillary force, which are available to manipulate microparticles or biological entities [15,16,17]. Hence, phenomena such as separation, enrichment, focusing and aligning, patterning, trapping, sorting, and isolation can be realized. Mixing is often required for sample dilution, reagent homogenization, and chemical or biological reactions [18]. However, due to its micron-scale dimension, the flow behavior inside the microchannels is laminar, and mixing is achieved through a long process of molecular diffusion. This issue can be resolved by creating chaotic advection or turbulence through active or passive micromixers [19] or the utilization of flexible and soft walled microchannels [20]. As to the substrate materials for the microfluidic chips, silicon, glass, polymers, metals, ceramics, paper, natural biomass polymer scaffolds [21,22,23,24,25], or a hybrid of the above have been utilized. Although silicon was the first material to be adopted due to its processability through semiconductor processing techniques, polymers have emerged to have their own place thanks to their numerous merits, such as chemical inertness, thermal stability, mechanical flexibility, optical transparency, biocompatibility, biodegradability, good electrical insulation, diverse surface modification with ease, recyclability, and low cost. In addition, the innovative strategies/methods that have been developed and are complementary to existing polymer manufacturing techniques smoothen the transition of microfluidic chips from lab test samples to market-mature products. As to the subsequent packaging or bonding of polymer microfluidic chips, techniques have been established relatively well by taking advantage of the characteristic physical properties of polymers (e.g., glass transition temperature (T_g_), higher impact strength, and easy surface modification). Even the application of screw clamps or devices fixed with screw nuts was reported for the assembly of microfluidic chips. It is worth mentioning that polydimethylsiloxane (PDMS), one of the polymeric materials, has become a standalone material, and techniques related to using PDMS are included in the domain of so-called soft lithography [26,27].

In this paper, different techniques for fabricating polymer microfluidic chips were reviewed. To create open microchannels on polymer substrates, the fabrication techniques are classified into two categories—that is, mold-based and non-mold-based approaches. The open microchannels are then sealed with polymers or other types of materials to form the closed microchannels through various bonding methods. Owing to the tremendous amount of work and reviews regarding this field, the discussion is limited to constructing microchannels on polymer substrates, and the literature is mainly focused on those published in the past 5 years. The fabrication techniques of polymer microfluidics discussed in this review are shown in Figure 1. For mold-based techniques, micro-embossing, micro-injection molding, and casting were reviewed. As for the non-mold-based techniques, CNC micromachining, laser micromachining, and 3D printing were reviewed. For the packaging or bonding of polymer microfluidic chips, several great reviews can be found in the literature [28,29]. Although cellulose-based filter paper is composed of polymers and could be counted as a polymeric material, paper-based microfluidics was not included in this review. A comprehensive review of paper-based microfluidics can be found elsewhere [30,31]. In addition, since the wood-based and the plant-based biotemplates can be directly used as the microfluidic platform without involving too much device fabrication, they are not included in this review either.

## 2. Mold-Based Techniques

### 2.1. Micro-Embossing

Micro-embossing is one of the most widely used techniques to fabricate polymer microfluidics thanks to its simple process, low tooling cost, no need for well-trained personnel, and affordable equipment. It is good for prototyping and can be extended and modified to become a continuous process (e.g., roll-to-roll, roll-to-plate, etc. [32]). Several reviews can be found in the literature [33,34,35]. In principle, a mold insert with the desired pattern of the microfluidics needs to be constructed first, which can be obtained through various methods ranging from lithographic technique, silicon dry/wet etching, laser ablation, electrodischarge machining (EDM), and CNC machining to 3D printing [36]. Once the mold is obtained, it is placed on top of the polymeric substrate, and the assembly is heated above the glass transition temperature of the polymer. The mold is then subjected to a compression force, allowing the pattern to be completely transferred onto the substrate. The assembly is cooled below the glass transition temperature (T_g_), and the substrate is separated from the mold. Depending on the mold and substrate temperatures, micro-embossing can be classified into isothermal and non-isothermal processes.

#### 2.1.1. Isothermal Micro-Embossing

For isothermal micro-embossing, both the mold and polymer substrate are maintained at the same temperature prior to applying the compression force. In most of the scenarios, the micro-embossing temperature is set approximately 10–30 °C above the T_g_ of the polymeric materials. A vacuum may or may not be applied during the micro-embossing process. The mold is then compressed against the polymer substrate and remains in contact under compression for a period of time. Subsequently, the temperature is decreased below the T_g_, and the polymer substrate is separated from the mold. When constructing the microfluidics on the polymer film, the gas-assisted micro-embossing or a similar process called microthermoforming can be applied [37]. In this method, the gas pressure was utilized to compress the polymer film to conform with the mold feature, and the final part with the protruded structures was usually obtained. Focke et al. applied the design of experiments to optimize the process where the PDMS mold was used to fabricate microchannels on cyclic olefin copolymer (COC). The results showed that the moldability was greatly affected by the molding temperature, and errors such as film rupture, wrinkles, and non-sharp edges could occur during the process. These errors were eliminated, and the 300 μm wide microchannels with a sharp edge and less than 10 μm variation between mold insert and replica were obtained after process optimization. In addition to setting the micro-embossing temperature above T_g_, micro-embossing at the temperature below T_g_ was also demonstrated [38]. This was achieved by utilizing the solvent to “soften” the polymer substrate, i.e., plasticizing the surface of the polymer substrate. In this study, the solvent was the CO_2_ gas. By dissolving CO_2_ in the polymethyl methacrylate (PMMA) substrate, the processing temperature and pressure were substantially reduced, leading to lower residual stress [38]. Although organic solvents such as trichloroethylene, chloroform, or toluene were used to soften the surface of the polymer substrate, the depth of the channels after embossing (or, more appropriately named, imprinting) was only in the nanometer range [39,40]. Micro-embossing at room temperature, or cold forging, was also performed on polytetrafluoroethylene (PTFE) to create microchannels [41]. Although the channel depth could be fully replicated from the mold insert provided high compression force was applied, the channel width became 30% smaller than that on the mold. This was alleviated by applying high forging speed and longer dwell time. Besides the common thermoplastic polymers [34], the biocompatible and biodegradable polymer, polycaprolactone was also micro-embossed to form microchannels with a minimum channel width and depth of 75 and 38 μm, respectively [42].

#### 2.1.2. Non-Isothermal Micro-Embossing

Non-isothermal micro-embossing is when the mold temperature is different from that of the polymeric substrate during the micro-embossing process. This was first carried out by Juang et al. to investigate the effect of non-isothermal processing conditions on the replication accuracy [43,44]. The mold temperature was set either higher or lower than that of the polymer substrate. Although the feature size was in the mm range, interesting flow behaviors were observed. When the mold temperature was lower than that of the polymer substrate, the polymer deformed and was squeezed outward, resulting in poor replication. On the other hand, when the mold temperature was much higher than the T_g_ of the polymer and the polymer substrate was at room temperature prior to embossing, the polymer at the surface and in contact with the mold melted rapidly. The molten polymer then conformed with the mold feature under compression and filled the mold cavity, resulting in great replication accuracy. Yao et al. further investigated the process [45] and subsequently developed a strategy to construct a mold with a rapid thermal response (RTR) to shorten the cycle time from the regularly around 10 min to 20 s [46]. Laser/IR was also utilized as the heating source in micro-embossing [47,48]. Lu et al. performed laser/IR-assisted embossing in two fashions, e.g., transparent mold embossing (TME) and transparent substrate embossing (TSE). For the former, the IR passed through the mold and heated the polymer substrate for subsequent embossing. For the latter, the IR passed through the polymer substrate and heated the micro-protrusion on the carbon-black filled mold, where it absorbed the IR energy, and heating occurred locally around the protrusion. The flow pattern of the polymer observed in experiments agreed well with the simulation results. Chen et al. developed an IR-assisted hot press system to emboss the PMMA substrate and bond the microchannels. The results showed that the relative standard deviations were 3.1, 2.8, and 4.3% for the bottom width (51.1 μm), the top width (112.2 μm), and the depth (37.4 μm) of the embossed channels, respectively, indicating satisfactory chip-to-chip reproducibility [48]. Ultrasonic vibration was applied in micro-embossing as well [49,50,51]. In principle, the sonotrode is first in contact with either the polymer pellets/substrate/film, which is on top of the mold, or the mold, which is on top of the polymer substrate/film. For the former, the oscillatory energy is dissipated through the polymer pellets/substrate/film, which is heated and melted to fill the cavities on the mold. For the latter, the energy is dissipated through the mold, and the polymer substrate in contact with the protrusion is rapidly heated, as shown in Figure 2 [51] (note that the mold can be placed at the bottom with the substrate in contact with the sonotrode). This leads to localized deformation, and the heating and cooling time only takes approximately a few seconds. Sucularli et al. investigated the process-affected zone during ultrasonic embossing [51]. The results showed that the process-affected zone was a half-circle around the cross-section of the embossed feature, which was bounded by the isothermal surface at the glass transition temperature. Microchannels 200 μm in width and 150 μm in depth were fabricated. Runge et al. utilized ultrasonic embossing to fabricate microfluidic devices on polycarbonate (PC) for yeast cultivation, and the cell viability was demonstrated [52]. Bavendiek et al. developed an improved calcification propensity test for the assessment of phosphate toxicity using PC-based microfluidic chips fabricated by ultrasonic embossing [53]. The results showed that the test time was substantially reduced at a higher, controlled operating temperatures.

### 2.2. Microinjection Molding (μIM)

#### 2.2.1. Conventional μIM

For conventional injection molding (IM), the plastic pellets or granules are fed into a heated barrel from a hopper, and a screw-type plunger rotates and moves the plastic material slowly forward. The polymeric material starts to melt as it moves forward, and the polymer melt accumulates in front of the plunger. This generates the pressure to push the plunger moving backward. As the designated amount of polymer melt is prepared, the plunger then stops rotating and advances to force the polymer melt through the nozzle that rests against the mold, allowing it to enter the mold cavity through a gate and runner system. The mold remains cold so that the polymer melt solidifies almost as soon as the mold is filled [54] and subsequently separates from the mold. For microinjection molding (μIM), however, it is not just a scaling down of the conventional injection process [54], and there are several technical issues, such as the mold construction technology, raw material variation, product properties, modeling of the molding process, etc. [55]. For example, LIGA-based techniques, which involve lithography, electroplating, and molding or μEDM, are necessary for making high-aspect-ratio mold inserts with tight dimensional tolerance. The separate plasticization and injection units are adopted in μIM for accurate melt metering, as shown in Figure 3 [56]. The mold temperature is raised higher than T_g_, and venting the high-temperature gas trapped in the cavities is critical to avoid the melt being stopped or burnt. The microscale rheological properties at a high shear rate, critical stress for wall slip, and so on are required for better process simulation [57,58]. The strong elastic effect, especially at the locations where an abrupt change of channel geometry occurs, and the onset of the flow instability due to channel dimension [59] need to be taken into consideration. Different polymers were used to make microdevices through microinjection molding, such as PMMA [60], PS [61,62,63,64], PP [64], and COC [64,65]. Lee et al. constructed microchannels with depths ranging from 150 to 500 μm, and the widths ranged from 200 to 700 μm on PS, PP, and COC, and microwells with 100 μm depth on COC and PP for cell culture [64]. Viehrig et al. constructed nanostructures on COC through injection molding, which were subsequently deposited with gold to form Au-capped polymer nanocones [65]. The device was used for SERS detection, and an enhancement factor of ~5 × 10^6^ with a relative standard deviation of 14% over the sensor area (2 × 2 mm^2^) and an 18% signal variation among substrates were achieved.

#### 2.2.2. Non-Conventional μIM

##### Variotherm

Unlike the conventional IM, where the mold temperature is kept constant or varied well below the melting temperature (T_m_) or T_g_ of the polymer during the injection molding cycle, the mold temperature in the variotherm process can be close to T_m_ or T_g_, and the variation of the mold temperature is tremendous, as shown in Figure 4 [66]. The rationale behind the variotherm process is to have a hot mold during the injection stage and a cold mold during the cooling stage with a rapid change of the mold temperature [67]. By doing so, the cycle time of μIM can be substantially reduced and the mold insert with microstructures, especially the high-aspect-ratio microstructures, can be completely filled. In order to achieve a rapid change in the mold temperature, several methods have been developed. For example, the mold can be convectively heated using hot air, oil, water, or steam. Radiation heating from various sources/strategies such as infrared lights/lamps, laser, the proximity effect from passing a high-frequency current through an electrical coil, and dielectric heating were applied to heat the mold insert. The electrically resistive heating, using heating cartridges or electric heating rod [68] to heat the mold insert through thermal conduction, was also utilized. Induction coils were used to provide induction heating as the result of the eddy current loss in the mold [69]. Comparisons between various heating methods can be found in the literature [67]. For instance, proximity heating does not have a significant influence on the mold lifetime, but the heating uniformity is not good, and it is not suitable to be used to heat the complex mold. Oil/steam heating is good for heating uniformity and is suitable to be used to heat the complex mold. However, the mold lifetime is greatly affected. Ultrasonic heating overall provides an acceptable heating source in terms of heating uniformity, heating complex mold, and less influence on the mold lifetime. In general, utilizing the variotherm process can reduce the cycle time and flow-induced molecular orientation, increase the flow path, minimize the weld lines, and improve the replication accuracy of high-aspect-ratio microstructures [67], but shorten the lifetime of the mold insert due to thermal fatigue. Zhang et al. applied variotherm to fabricate a microlens array on COC, where the radius of the sphere was 419.5 μm and the pitch was 190 μm [70]. By using the variotherm process, the general residual stress level and uniformity of the microlens array area were improved by 5.08% and 88.11%, respectively. Zhang et al. investigated the replication integrity of micro features made by the variotherm process [71]. Microfluidic channels with wave and droplet patterns and micro features with square and ellipse patterns were examined. With optimization, 100% replication was achieved, and micro features with aspect ratios up to 10 were obtained.

##### Ultrasonic-Assisted μIM

In recent years, ultrasound has been incorporated in μIM where material degradation and waste are most important to be considered and need to be avoided [50]. Since the sonotrode provides the oscillatory energy where the mechanical energy is transformed into thermal energy, it dissipates through the polymeric material that the sonotrode is in contact with, and the polymer pellets are heated and melted. In doing so, the merits are three-fold. One is that heating the barrel is not required, leading to a saving of energy [72]. Another is that only the required amount of raw material is ultrasonically plasticized, and thus the waste is minimized. The other is that the molten polymer can rapidly fill the mold cavity such that the residence time is reduced and thermal degradation is prevented. For ultrasound-assisted μIM, it can be divided into two types. That is, ultrasound-assisted micro-injection molding (UAMIM) and ultrasonic plasticization micro-injection molding (UPMIM), as shown in Figure 5 [73]. For the former, the ultrasonic vibration is applied to the molten polymer in the mold cavity such that the fluidity of the molten polymer and molding quality of micro-parts are enhanced [74]. Application of ultrasonic vibration can have different settings, such as being direct or indirect contact with the molten polymer, vertical or parallel to the melt flow direction, and part of or the whole mold. As to the latter, ultrasonic vibration is applied to plasticize the polymer raw materials. Two types were developed, with one featuring “injection while plasticizing” and the other “injection after plasticizing”. The former is further divided into two categories: one has the moving sonotrode, and the other has the moving plunger. Besides energy saving, complete filling of the mold cavity, and less waste of raw material, other benefits might come along with using ultrasound, such as improved fluidity of the molten polymer and reduced residence time.

### 2.3. Casting

Casting has been the most common, widely used technique to fabricate microfluidic chips due to its low capital cost, simple procedure, and high fidelity [26]. Since the pioneered work carried out by Whitesides’ group, PDMS has been the primary material used in casting, and the techniques derived from using PDMS are referred to as soft lithography [27]. Besides the abovementioned merits, the chemical and optical properties of PDMS make them attractive to be the substrate for microfluidics. They are optically transparent; chemically inert; nontoxic; biocompatible; gas-permeable; have a low autofluorescent background; and have easy surface modification through plasma, UV, or silanization. To fabricate microfluidic chips through casting, the first step is to construct a mold with the desired microfluidic design with protruded structures (the positive mold). Once the mold insert is obtained, the PDMS base and curing agent are mixed at the weight ratio of 10:1 and degassed, followed by being poured onto the mold. The assembly is then placed either at ambient condition for at least 24 h or in the oven set at a certain temperature (e.g., 65 °C) for a period of time (e.g., 4 h). The PDMS replica with microchannels is then separated from the mold. Note that the PDMS replica can be used to produce the positive mold by casting epoxy or molding with plastics (e.g., PMMA, PC [75]) such that the mother mold can be preserved for a longer time. If the structures on the mold are not protruded (i.e., the negative mold), then double-casting may be applied to obtain the PDMS microchannels [76]. This is achieved by first casting the PDMS onto the negative mold to obtain the positive PDMS mold. The positive PDMS mold is then either treated with a mold releasing agent, modified with Teflon [77], PEG [78], or subjected to thermal aging [79]. The PDMS is subsequently cast onto the treated positive PDMS mold to obtain the PDMS replica with microchannels.

## 3. Non-Mold-Based Techniques

### 3.1. CNC Micromachining

For rapid prototyping of microfluidic chips with a channel dimension larger than 100 μm, modern CNC micromachining will always be an option to be considered due to its fast turnaround time, simplicity, no need for cleanroom facilities, and low cost of equipment. In addition, structures with high aspect ratios can be easily manufactured [80]. Since the speed and position of the endmills are controlled by CNC, the computer-aided design (CAD) or computer-aided manufacturing (CAM) files can be used directly to send instructions to the motor and servos. To achieve low surface roughness and burr formation, the operating conditions, such as selection of endmills, spindle speed, feed rate, and axial and radial runout, need to be thoroughly considered. Note that when fabricating microchannels with dimensions less than 100 μm, handling the endmills requires extreme caution to prevent its breakage [36]. Lashkaripour et al. fabricated a microfluidic droplet generator through CNC micromilling. The minimum feature size was 75 μm, and the surface roughness of the microchannels was approximately 0.205 μm. A general procedure to calculate feed rate and spindle speed for any sub-millimeter endmill was provided, and the surface quality was mainly determined by stepover [81]. The flow-based microfluidic gradient mixer was also fabricated by three-axis CNC micromilling on the stressed polystyrene sheet, and the laminar flow pattern was demonstrated [82]. Utilization of the endmill with a diameter less than 50 μm was demonstrated, and low surface roughness and burr formation were achieved [83].

### 3.2. Laser Micromachining

Laser micromachining involves laser milling, laser drilling, laser cutting, laser etching, and laser engraving, which share the common phenomenon of laser ablation. Usually, a high-energy laser beam is used and focused on the substrate surface, where the materials are removed due to melting or vaporization as a result of absorbing the photon energy. The transition from solid to gas leads to the formation of a plasma plume, as shown in Figure 6 [84]. During this transition, the molten polymer experiences the explosive liquid–vapor phase transition as the temperature at the laser–substrate interaction zone further increases and, thus, the material is removed. The laser sources used in laser micromachining can be classified based on their wavelengths (UV/excimer lasers and infrared lasers) or the time scale of their pulse durations (millisecond, microsecond, nanosecond, picosecond, and femtosecond lasers). Laser micromachining possesses several advantages over other traditional machining techniques, such as being precise, fast, clean, contactless, flexible, not having tool wear, and reducing industrial effluents [85,86,87,88]. Many polymeric materials have been used in laser micromachining [89], such as PMMA, COP, PS, PC, PET, biodegradable polymers [90,91], and PDMS. For example, Genna et al. applied laser micromachining on PMMA, and it was found that a surface with low roughness was obtained by operating the 30 W RF excited CO_2_ laser with a fundamental wavelength of 10,600 nm at high scanning speed and continuous wave mode [92]. Ghoochani et al. constructed microchannels on polyether sulfone (PES), and a model was developed to predict the depth of the microchannels with an average error of 8.1% [93]. Liu et al. fabricated microchannels on cyclo-olefin polymers, and it was found that the microchannels with a Gaussian-like profile were obtained. In addition, the channel width and depth increased as the laser power increased but decreased as the scan speed increased [94]. Gao et al. utilized a diode laser to construct microchannels on PMMA [95]. It was concluded that microchannels with a semi-circle cross section were obtained where the channel width and depth ranged from 300 to 500 μm and 50 to 130 μm, respectively. Yin applied CO_2_ laser ablation on PET to obtain microchannels, and the channel width ranged from 165 to 315 μm, and the depth ranged from 20 to 250 μm [96]. Vargas et al. used a continuous-wave CO_2_ laser to fabricate microchannels on PMMA [97]. The channel width varied from 58 to 264 μm, and depths ranged from 14 to 136 μm. The standard deviation was lowest for a scan speed of 40–100 mm/s at powers between 1.8 and 3 W. Fan et al. performed CO_2_ ablation on PS to create microchannels where the channel width and depth ranged from 60 to 140 μm and from 20 to 150 μm, respectively [98]. If the biaxially oriented PS was used, the range of width ranged from 25 to 60 μm, and the depth ranged from 100 to 800 μm. Chen and Hu used a CO_2_ laser to fabricate microchannels on a PC, and the results showed that the optimal parameters were an 8 W laser power and 15 mm/s scan speed for making a 70 mm-long channel [99]. Hsieh et al. utilized an excimer laser to fabricate microchannels on PDMS and biodegradable polymers, poly(glycerol sebacate) (PGS), and poly(1,3-diamino-2-hydroxypropane-co-polyolsebacate) (APS) [87]. The results showed that the width of the microchannel was correlated to the beam size. In addition, increasing the beam size and the number of repeated scanning increased the depth of the microchannels, where PGS increased the most among the three polymers. Min et al. performed successive laser pyrolysis on PDMS, which was coated on the glass substrate in advance [100]. The pyrolyzed SiC was detached from the glass substrate to create a PDMS microchannel. The hemispherically shaped cross-section of the microchannel was obtained where the widths of the uppermost surface and the interface between PDMS and glass were 115 to 120 μm and 7 to 8 μm, respectively. Scanning four times expanded the width of the PDMS–glass interface to 62 μm.

### 3.3. 3D Printing

In recent years, 3D printing, an additive manufacturing technique, has been receiving great attention for the fabrication of microfluidics due to its advantageous features, such as no cleanroom required, the low cost of consumables and equipment, great accessibility, fast production, easy edition and reprint of designs and multi-materials, and multiphase printing. Moreover, robust connection ports and complex flow regulating components can be constructed, and the integration of detectors and cell culture on chips can be achieved [101]. Because of these features, several issues related to conventional techniques could be resolved—for example, expensive and time-consuming processes when changing the device designs and difficulty in transitioning from prototyping fabrication to bulk manufacturing [102,103,104,105]. There are different ways to realize 3D printing, such as stereolithography (SL) [106], selective laser melting and sintering (SLS), fused deposition modeling (FDM), and polyjet or multi-jet modeling (MJM), as shown in Figure 7. For SL, the photosensitive thermoset polymers in the liquid form are selectively, layer-by-layer, repetitively photopolymerized through UV light, a high-intensity laser, or focused LED light sources to form the 3D structures. Using the digital light projection (DLP) technique, an entire layer of resin can be exposed at once. Two strategies are exploited, e.g., free surface configuration, where photopolymerization occurs at the top surface of the vat, and bat configuration, where it occurs at the bottom surface. For SLS, the 3D structure is composed of layers of fused powder materials. By using a focused laser beam, the powder materials are heated and fused together through chemical reactions, solid-state sintering, or melting. The process is repeated for each layer, and the 3D structure is constructed. For FDM, the 3D structure is completed through a layer-by-layer process. One layer at a time is created by extruding the polymer materials from a heated nozzle, which cools and hardens immediately after extrusion. The process is then repeated, and multiple layers are constructed to form the structures. For MJM, the photopolymer droplets are delivered through multiple inkjet heads integrated with a movable platform, and the droplets are cured rapidly by UV light to form a layer, i.e., a slice of the 3D structure. Materials with various properties and different colors can be used. The process is then repeated, and the construction of the entire structure is achieved layer-by-layer. Comparisons of these approaches can be found in the literature [107]. There are examples using 3D printing techniques. Piironen et al. fabricated microchannels through SL using four SL resins to investigate cell adhesion and proliferation [108]. The microchannels were 34 mm long and tapered with a linearly decreasing width (from 500 to 300 μm) and height (from 1050 to 150 μm), which can be used to determine the shear force threshold. The results indicated that Dental SG was the most favorable resin of the four resins for microfluidic organ-on-a-chip applications and cell cultivation. Kamperman et al. utilized SL to construct 3D parallelized microfluidic droplet generators where the channels with radially multiplexed designs were stacked [109]. The results showed that microchannels with a minimum channel dimension of 50 μm were fabricated, and equal per-nozzle flow rates were achieved, which offers a strategy to increase throughput while maintaining a smaller device footprint. Mader et al. applied FDM to construct PS microchannels for cell culture experiments [110]. The minimum channel dimension was 300 μm, and the enclosed microchannels with an aspect ratio of 10 were obtained. Sagging of the bridging layers would occur if the aspect ratio was less than 0.2. Kotz et al. fabricated microchannels on PMMA through FDM for mixing and biofunctionalization tests [111]. The minimum channel dimension was 300 μm, and microchannels with microstructures embedded inside were also obtained. Nelson et al. constructed microchannels on polyurethane (PU), and the minimum channel dimension was 50 μm [112]. Sagging of the bridging layer was minimized by the shape of the ceiling geometry, and microchannels with different cross-sectional shapes, such as square, triangle, and ellipsoid, were obtained. It was found that the surface roughness of the microchannel could be reduced when the nozzle was brought closer to the print surface. Sweet et al. constructed concentration gradient generators through MJM to identify optimal drug compositions [113]. The generators were a 3D microchannel network with 3 inlets and 13 outlets, as shown in Figure 8. The symmetric concentration gradient for three fluids was generated, and the cocktail solutions for bacteriological studies were carried out.

## 4. Conclusions

In this article, fabrication techniques of polymer microfluidics were reviewed and compared as shown in Table 1. Since there are pros and cons for each method, one should select the proper technique to meet his/her own purposes. For example, if the microfluidic chips are mature and ready for the market, then micro-injection molding with high-throughput capability will be able to meet market demand. As for the proof-of-concept, PDMS casting is the choice to make in terms of a simple procedure and low cost. If thermoplastic polymers are the substrate materials for the microfluidic chips, micro-embossing, CNC micromachining, and laser micromachining are the options where micro-embossing provides microfluidic chips with lower surface roughness and a sufficient number of samples in a relatively short time for repetitive experiments. Microfluidic chips with complex and 3D designs can be fabricated by CNC micromachining, laser micromachining, or 3D printing processes. These methods can also be utilized to create mold inserts for mold-based techniques. It is worthy of mentioning that it would be better if the polymeric materials used in prototyping were the same as or similar to those which can be processed in mass manufacturing. Doing so could minimize the obstacles in the transition from the lab samples to the market products. Among the three prototyping techniques mentioned previously, 3D printing is highly praised and appreciated due to its great versatility. Since limited printing materials were mentioned by the researchers, the development of printing materials for 3D printing that can be used for mass manufacturing is imperative. Another is that the development of 3D printing equipment with a resolution of approximately 10 μm and a lower price (e.g., around $ 15,000) would further accelerate research and development in polymer microfluidics. This would mean at least a five-times decrease in the equipment cost. Moreover, it would also imply less utilization of CNC micromachining, laser micromachining, and even photolithography when prototyping polymer microfluidics.

## Figures and Tables

**Figure 1 polymers-14-02028-f001:**
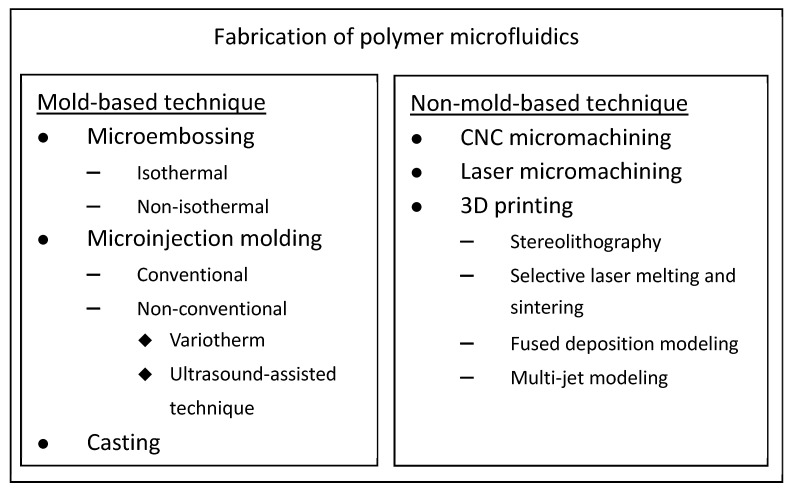
Fabrication techniques for polymer microfluidics.

**Figure 2 polymers-14-02028-f002:**
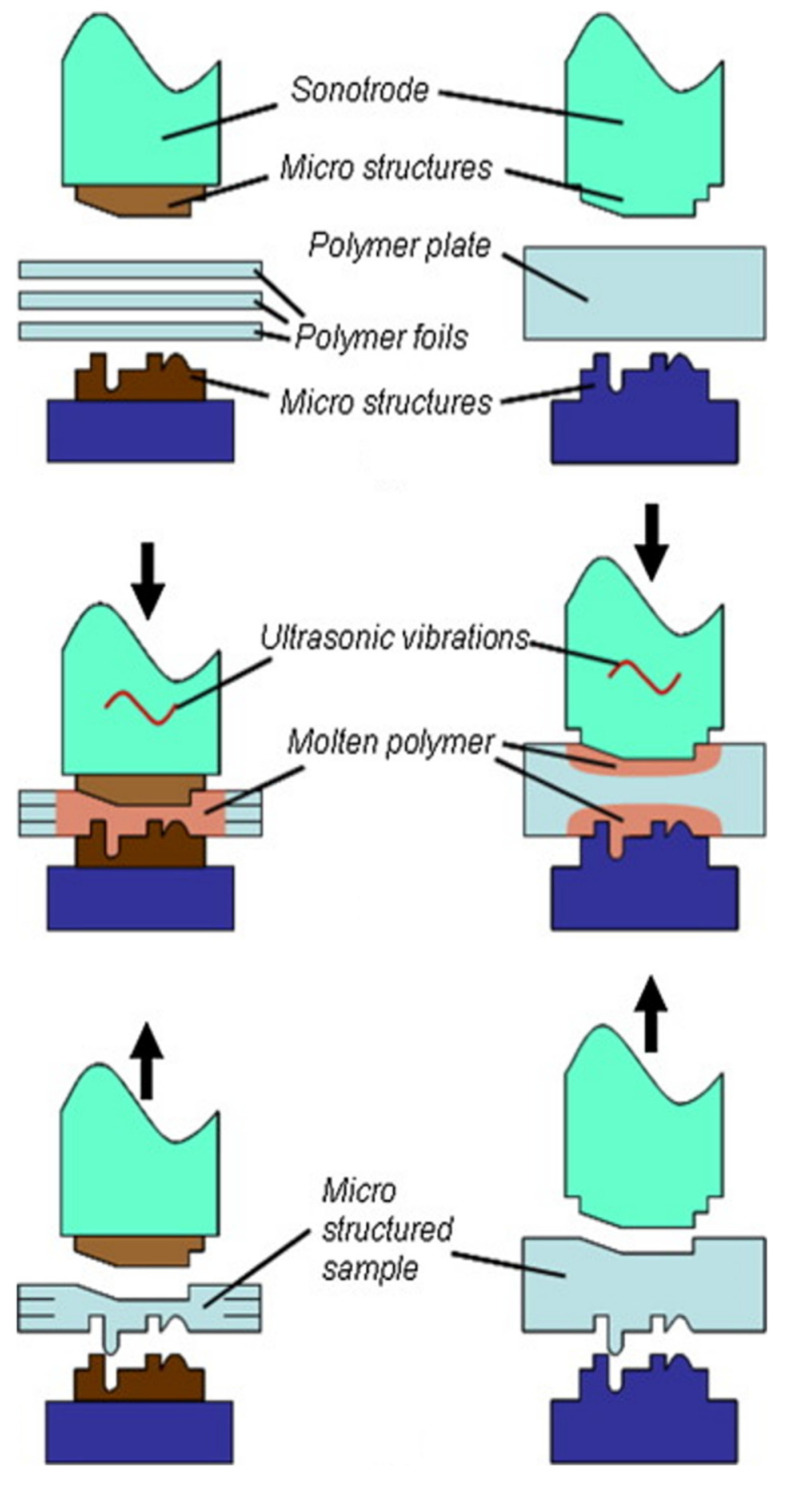
Ultrasonic hot embossing of a stack of foils (left) and a plate (right). Adapted, with permission, from [51] 2015, Elsevier.

**Figure 3 polymers-14-02028-f003:**
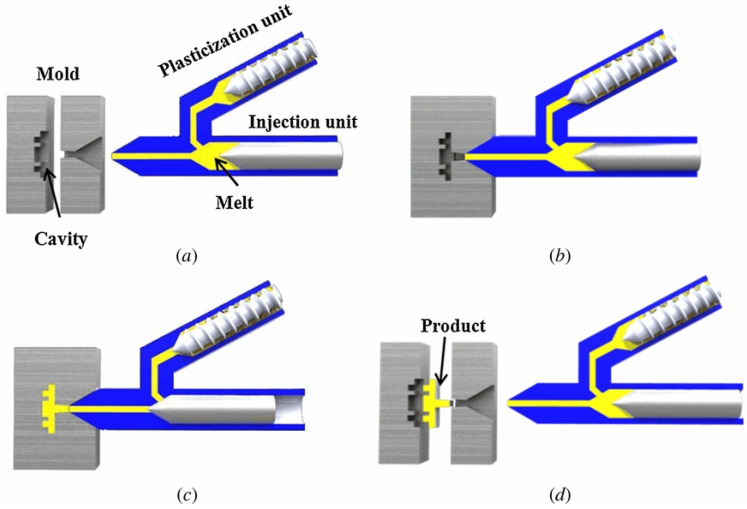
μIM process demonstration with separate plasticization and injection units: (**a**) plasticization, (**b**) mold closing, (**c**) injection, packing and cooling, and (**d**) demolding and re-plasticization for the next cycle. Adapted, with permission, from [56]. 2013, IOPscience.

**Figure 4 polymers-14-02028-f004:**
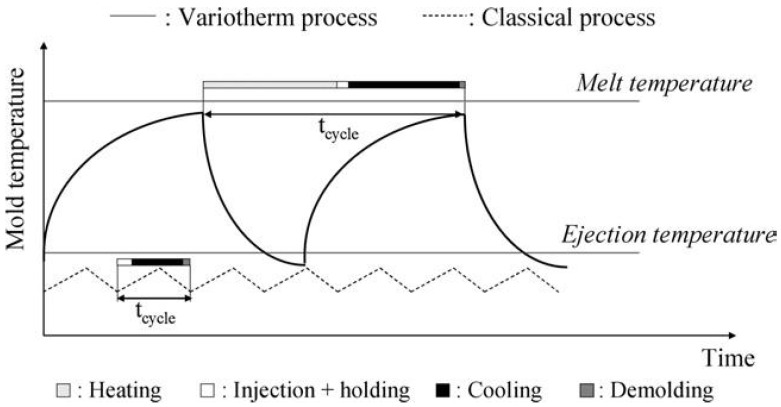
Comparison of mold temperature in the classical and variotherm processes. Adapted, with permission, from [54]. 2007, IOPscience.

**Figure 5 polymers-14-02028-f005:**
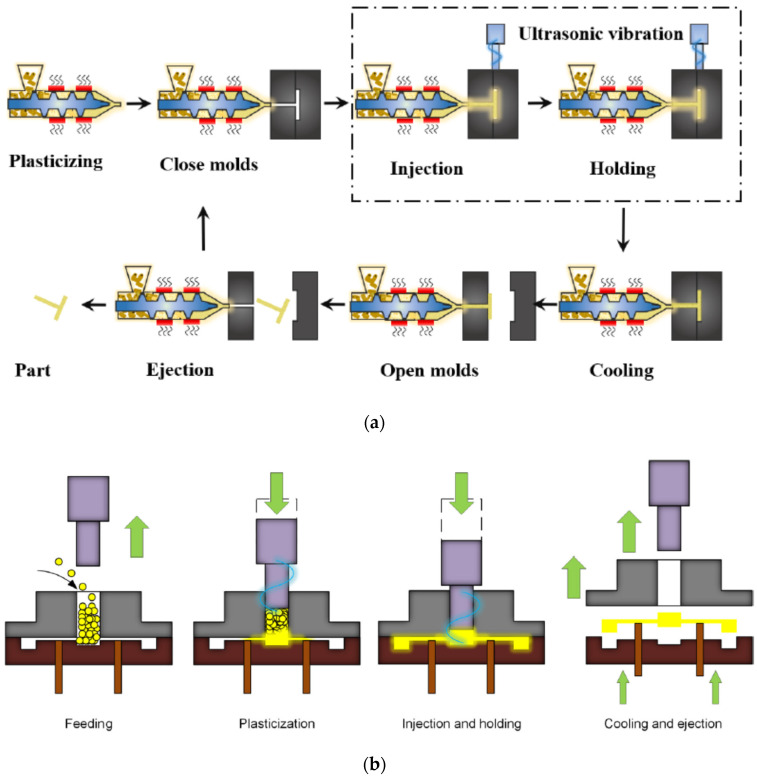
Schematic diagram of (**a**) UAMIM and (**b**) UPMIM. Adapted, with permission, from [73]. 2021, MDPI AG.

**Figure 6 polymers-14-02028-f006:**
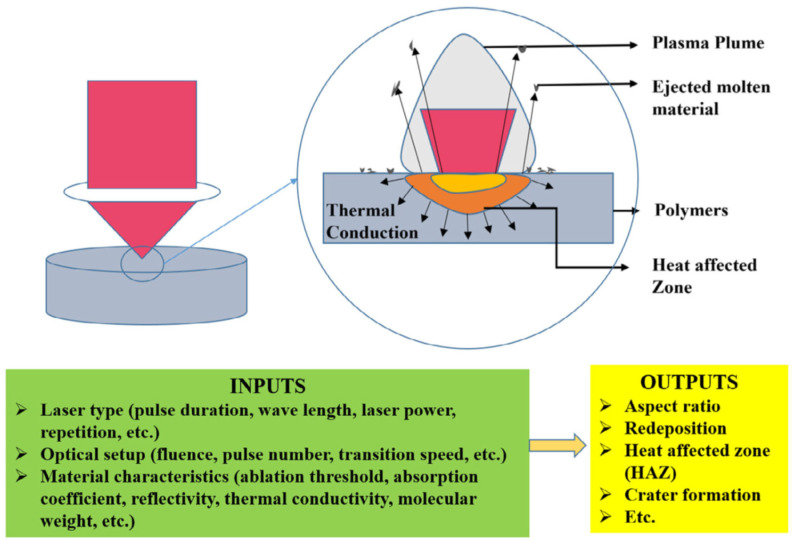
Mechanism at the laser–material interface. Adapted, with permission, from [84]. 2019, Elsevier.

**Figure 7 polymers-14-02028-f007:**
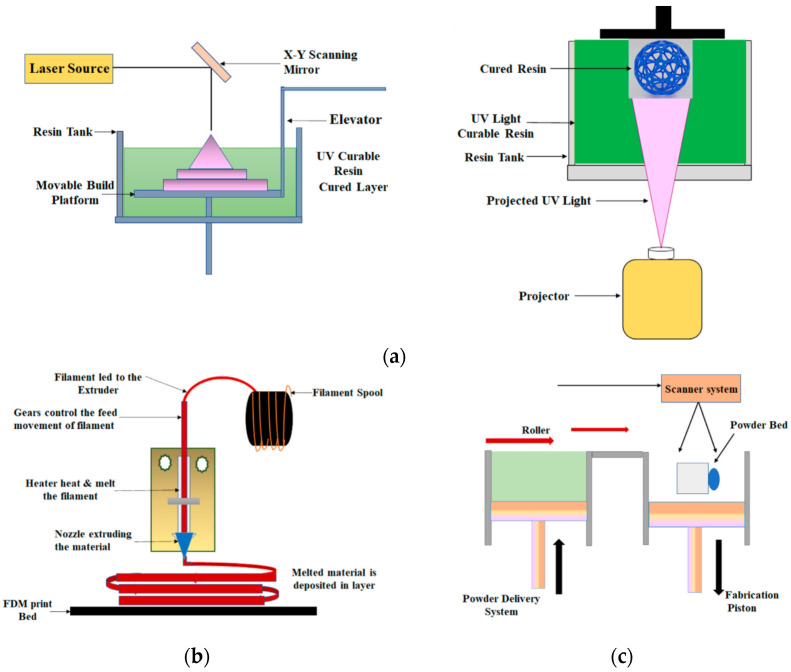
Schematic showing the setup for (**a**) stereolithography, (**b**) fused deposition modeling, and (**c**) selective laser sintering. Adapted, with permission, from [105]. 2021, Frontiers.

**Figure 8 polymers-14-02028-f008:**
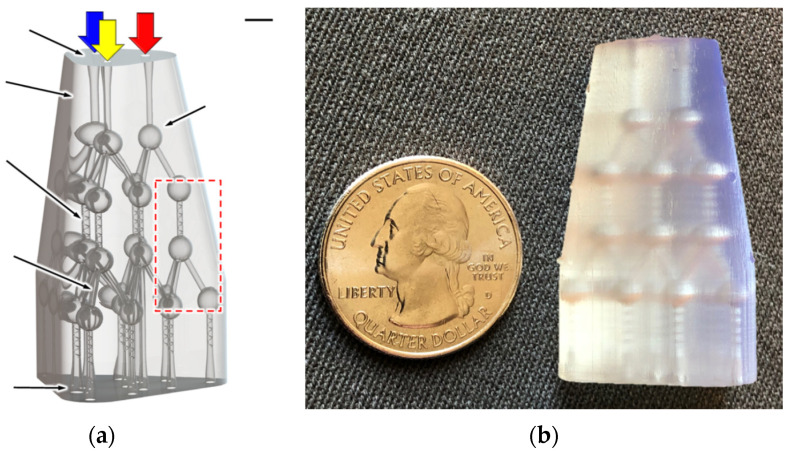
Developed 3D μ-CGG prototype and experimental setup. (**a**) Reverse solids model, representing the manufacturable 3D microfluidic design comprising a single solid body with embedded hollow microchannel structures. (**b**) Fabrication results, actual 3D μ-CGG prototype after postprocessing; hollow interior structures are visible through the semitranslucent structural material, with US quarter for scale. Adapted, with permission, from [113]. 2020, Spring Nature.

**Table 1 polymers-14-02028-t001:** Comparison of fabrication techniques for polymer microfluidics.

	Processing Speed	Throughput	Equipment Cost	Complexity in Fluidic Design	Working Materials	Resolution
Micro injection molding	5	5	5	2	3	5
Micro-embossing	4	4	2	2	3	5
PDMS casting	1	1	1	2	1	5
CNC micromachining	3	3	2	3	5	3
Laser micromachining	3	3	3	3	5	3
3D printing	2	2	3	5	3	2

Five-level scale: 5 refers to high, fast or many; 1 refers to low, slow or few.

## Data Availability

Not applicable.

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
