# Peer review of "Fabrication of Polymer Microfluidics: An Overview"

_polymers, 2022, doi:10.3390/polym14102028_

Round 1

Reviewer 1 Report

In this manuscript, the primary fabrication techniques for polymer microfluidics were reviewed. This is an important work. I suggest to publish this paper after revision.

Some suggestions can be found in the annex.

Reviewer 2 Report

Recommendation: Publish as is, or with minor revisions as noted.

Comments:

            This is a nice review that covers the fabrication of microfluidics using polymeric materials. The authors divided the review into two major categories, mold-based, and non-mold-based approaches, and in each category, it was discussed in detail through several subcategories. In my opinion, this work can be published in Polymers after the authors address some questions below.

  • In the introduction part paragraph 1, references should be added after this sentence “In addition, there are various forces like… microparticles or biological entities”.

  • References should also be added to each of the techniques provided in Table 1.

  • This review article covers the fabrication of microfluidics, but it will be helpful to the readership if one or two paragraphs can be added regarding the application of microfluidics, such in the in-situ/operando characterization of catalysts or chemical reactions.

  • Turbulence was an important issue in microfluidics, especially in the design of microfluidics channels but not covered in this review. It will be helpful if the authors can talk about this to make the review article a comprehensive one for the audience of Polymers.

Author Response

This is a nice review that covers the fabrication of microfluidics using polymeric materials. The authors divided the review into two major categories, mold-based, and non-mold-based approaches, and in each category, it was discussed in detail through several subcategories. In my opinion, this work can be published in Polymers after the authors address some questions below.

  • In the introduction part paragraph 1, references should be added after this sentence “In addition, there are various forces like… microparticles or biological entities”.

ANS: The references were added.

  • References should also be added to each of the techniques provided in Table 1.

ANS: Table 1 was modified based on reference #115 along with authors’ perspectives.

  • This review article covers the fabrication of microfluidics, but it will be helpful to the readership if one or two paragraphs can be added regarding the application of microfluidics, such in the in-situ/operando characterization of catalysts or chemical reactions.

ANS: The reference regarding synthesis and characterization of catalysts was added.

  • Turbulence was an important issue in microfluidics, especially in the design of microfluidics channels but not covered in this review. It will be helpful if the authors can talk about this to make the review article a comprehensive one for the audience of Polymers.

ANS: A paragraph regarding turbulence in microfluidics was added.